# Incidence of Air Leaks in Critically Ill Patients with Acute Hypoxemic Respiratory Failure Due to COVID-19

**DOI:** 10.3390/diagnostics13061156

**Published:** 2023-03-17

**Authors:** Robin L. Goossen, Mariëlle Verboom, Mariëlle Blacha, Illaa Smesseim, Ludo F. M. Beenen, David M. P. van Meenen, Frederique Paulus, Marcus J. Schultz

**Affiliations:** 1Department of Intensive Care, Amsterdam University Medical Centers, Location ‘AMC’, 1105 AZ Amsterdam, The Netherlands; 2Department of Thoracic Oncology, Antoni van Leeuwenhoek Ziekenhuis, 1066 CX Amsterdam, The Netherlands; 3Department of Radiology, Amsterdam University Medical Centers, Location ‘AMC’, 1105 AZ Amsterdam, The Netherlands; 4Department of Anesthesiology, Amsterdam University Medical Centers, Location ‘AMC’, 1105 AZ Amsterdam, The Netherlands; 5ACHIEVE, Centre of Applied Research, Faculty of Health, Amsterdam University of Applied Sciences, 1091 GC Amsterdam, The Netherlands; 6Mahidol–Oxford Tropical Medicine Research Unit (MORU), Mahidol University, Bangkok 10400, Thailand; 7Nuffield Department of Medicine, University of Oxford, Oxford OX3 7BN, UK

**Keywords:** acute hypoxemic respiratory failure, acute respiratory failure, COVID-19, ARDS, positive pressure ventilation, invasive ventilation, high-flow nasal oxygen, barotrauma, air leaks, subcutaneous emphysema, pneumothorax, pneumomediastinum, chest tube

## Abstract

Subcutaneous emphysema, pneumothorax and pneumomediastinum are well-known complications of invasive ventilation in patients with acute hypoxemic respiratory failure. We determined the incidences of air leaks that were visible on available chest images in a cohort of critically ill patients with acute hypoxemic respiratory failure due to coronavirus disease of 2019 (COVID-19) in a single-center cohort in the Netherlands. A total of 712 chest images from 154 patients were re-evaluated by a multidisciplinary team of independent assessors; there was a median of three (2–5) chest radiographs and a median of one (1–2) chest CT scans per patient. The incidences of subcutaneous emphysema, pneumothoraxes and pneumomediastinum present in 13 patients (8.4%) were 4.5%, 4.5%, and 3.9%. The median first day of the presence of an air leak was 18 (2–21) days after arrival in the ICU and 18 (9–22)days after the start of invasive ventilation. We conclude that the incidence of air leaks was high in this cohort of COVID-19 patients, but it was fairly comparable with what was previously reported in patients with acute hypoxemic respiratory failure in the pre-COVID-19 era.

## 1. Introduction

Intensive care unit (ICU) patients in general, and mechanically ventilated, critically ill patients in particular, are at risk of air leaks [1]. ICU patients often need a central venous catheter, e.g., for continuous infusion of vasoactive agents and fluids, and the insertion of such catheters can be complicated by pulmonary perforations [2]. Critically ill patients receiving positive pressure ventilation are at an additional risk of barotrauma, e.g., when the ventilation needs to be applied at high volumes or pressures [3], or as a consequence of underlying lung disease [4].

Air leaks such as subcutaneous emphysema, pneumothorax and pneumomediastinum are frequently reported in patients with acute respiratory distress syndrome (ARDS) [5,6,7,8]. Recent literature suggests an increased incidence of air leaks in patients with ARDS due to COVID-19, as compared to patients with classic ARDS [9,10,11,12,13]. The unprecedented large numbers of critically ill COVID-19 patients early in the pandemic, and the strong interest of journals to publish anything that could be of interest, may have led to publication bias. Numerous reports or small series were published on radiographic findings in COVID-19 patients, including air leaks. On the other hand, it could also be that the incidence was, indeed, increased, which may have consequences for the way such patients should be monitored and treated

We wished to accurately determine the incidence of air leaks in critically ill COVID-19 patients. In order to do so, we re-scored the chest images of all COVID-19 patients who were included in two nationwide observational studies of respiratory support in the first year of the national outbreak in the Netherlands. The data were from patients with acute hypoxemic respiratory failure in one university-affiliated hospital. We also determined how many pneumothoraxes were treated with a chest tube. We hypothesized the incidence of air leaks in COVID-19 patients with acute hypoxemic respiratory failure to be high but comparable to what had previously been reported in patients with classic ARDS.

## 2. Materials and Methods

### 2.1. Study Design, Ethical Approval and Patients

This is a preplanned analysis of patients admitted to the ICU of the Amsterdam University Medical Centers (AUMC), location ‘AMC’, Amsterdam, The Netherlands, in two sequential observational studies, named ‘Practice of Ventilation in COVID-19 patients’ (PRoVENT–COVID) [14], and ‘Practice of Adjunctive Therapies in COVID-19′ (PRoAcT–COVID) [15]. The first study included patients in the first wave of the national outbreak, from 1 March to 1 June 2020, while the second study included patients in the second wave, from 1 October 2020 to 1 January 2021. The study protocols of the parent investigations were approved by the institutional review boards of the participating hospitals (W20_157 #20.171 and W20_526 #20.583), which waived the need for individual informed consent. Both studies are registered at clinicaltrials.gov (with the study identifiers NCT04346342 and NCT04719182).

Patients were eligible for participation in the two parent studies if: (1) aged > 18 years; (2) admitted to one of the ICUs of the participating hospitals; and (3) receiving invasive ventilation or high-flow nasal oxygen (HFNO) for COVID-19 that was confirmed by RT-PCR for SARS-CoV-2. PRoVENT–COVID and PRoAcT–COVID had no exclusion criteria. For the current analysis, we included only patients who received chest imaging during their admission to the ICU of our hospital.

### 2.2. Collected Data

On patients’ admission, the following baseline demographics were collected: sex, age, weight and height, and comorbidities, including chronic obstructive pulmonary disease, asthma, lung emphysema, interstitial lung disease, pulmonary cancer, history of pneumothorax and smoking status, which was defined as ‘smoking, a history of smoking, or never smoked’. We collected information about the type of respiratory support, which could be invasive ventilation or HFNO, its settings, and the Sequential Organ Failure Assessment (SOFA) score for disease severity. Radiology reports of images obtained during each patient’s stay in the ICU were screened for any mention of air leaks, i.e., subcutaneous emphysema, pneumothorax and pneumomediastinum.

Data were collected until day 90, with a check on day 28, in order to collect the date at which patients were weaned from respiratory support, the date of the ICU and hospital discharge and life status at the ICU upon hospital discharge.

### 2.3. Re-Assessment of Chest Images

All chest radiographs (CXRs) and chest computed tomography (CT) scans performed from ICU admission up to day 28, ICU discharge or death, if this occurred within 4 weeks, were retrieved and assessed for the presence of subcutaneous emphysema, pneumothoraxes and pneumomediastinum. A pneumothorax was scored if it was right-sided, left-sided or bilateral, and pneumomediastinum was scored if it was apical or basal. All CXRs and chest CT scans were independently scored by one trained medical student (investigator M.V.), one pulmonologist (investigator I.S.), and one pulmonologist-intensivist (investigator M.B.). If there was a disagreement, it was solved by reviewing the film together. In cases in which a consensus could not be reached, an independent radiologist (investigator L.B.) gave the final verdict.

Then, for each patient with pneumothorax on the CXR or chest CT scan, it was scored if a chest tube was used for drainage, and it was scored if the clinical file mentioned a tension component or hemodynamic instability due to the air leak. In any case of an air leak, the clinical file was screened for procedures that could have caused the air leak, including the placement of a central venous catheter or a tracheotomy.

### 2.4. Endpoints

The primary endpoint was the presence of an air leak—for this we used a collapsed composite of subcutaneous emphysema, pneumothorax or pneumomediastinum. Secondary endpoints were the type and location of air leaks, whether a chest tube was placed, and the duration of ventilation, length of ICU stay, and 28- and 90-day mortality.

### 2.5. Sample Size

We did not perform a formal power calculation for sample size; instead, the number of available patients served as the sample size.

### 2.6. Statistical Analysis

Categorical variables were presented as numbers and proportions. Continuous variables were presented as either means with standard deviations or medians with interquartile ranges where appropriate. Patients who developed air leaks were compared to patients who did not develop air leaks with respect to baseline parameters and ventilation characteristics. The groups were compared using a *t*-test and, where appropriate, a chi-squared test. The interobserver agreement between the medical student, the pulmonologist, and the pulmonologist-intensivist was calculated, and the frequency of consulting the study radiologist was reported.

Main ventilator parameters, including tidal volume (VT), positive end-expiratory pressure (PEEP), maximum airway pressure (Pmax) and respiratory system compliance (Crs), were presented in distribution plots, wherein vertical dotted lines represented the median for each variable, and the horizontal dotted lines represented the median of patients reaching each cut-off. Groups were compared using a t-test.

The incidence of the primary endpoint was reported as the percentage of patients who received imaging during ICU admission and had an air leak. If a patient received chest imaging more than once during ICU admission, the air leak was only scored the first time it appeared. The timing of air leaks was also visualized in a time to event graph. We also compared our findings to the radiology reports to discover if we found air leaks more often than reported in the final reports of the radiology department.

The duration of respiratory support, length of the ICU stay and life status upon discharge from the ICU, as well as the date at day 28, were calculated using the original dates. The length of time until liberation from respiratory support and the length of the ICU stay, respectively, were presented in plots, with death as a competing risk until day 90. The time to the first event was shown in a Kaplan–Meier curve.

Statistical analyses were performed using Rstudio (4.0.3). Statistical significance was set at 0.05.

## 3. Results

### 3.1. Patients 

Of a total of 2316 patients included in the two nationwide studies, 165 were admitted to the ICU of our hospital (Figure 1). Main reasons for exclusion from the two parent studies were an alternate diagnosis as the main reason for ICU admission or having received ventilatory support other than invasive ventilation or HFNO. Eleven (6.7%) patients were excluded from the current analysis due to missing chest imaging. The remaining 154 patients were predominantly male, with a high frequency of having a history of chronic obstructive pulmonary disease or asthma (Table 1).

There were neither differences in baseline characteristics nor in SOFA scores between patients with and patients without an air leak. Patients with an air leak were ventilated with a higher FiO2, but other ventilation characteristics were not different between patients with and patients without an air leak (Figure 2).

### 3.2. Incidence, Timing and Treatment of Air Leaks

A total of 712 chest images were available and re-assessed, revealing 542 CXRs and 170 CT scans, corresponding to a median of three (2–5) CXRs and a median of one (1–2) chest CT scan per patient. In 13 patients (8.4%), an air leak was identified on the CXR or the chest CT scan—this included subcutaneous emphysema, a pneumothorax or pneumomediastinum in seven (4.5%), eight (4.5%) and six (3.9%) patients, respectively. The locations of the pneumothoraxes and the occurrences of an isolated air leak—i.e., the presence of subcutaneous emphysema, a pneumothorax or a pneumomediastinum, alone or in combination—are presented in Table 2. Nearly half of the patients had a combination of air leaks.

In all of the chest imaging assessments, there was 100% agreement between the trained medical student, the pulmonologist, and the pulmonologist-intensivist—the study radiologist was never consulted. Our findings regarding air leaks were in line with what was reported in the radiology reports of all patients, i.e., when we found subcutaneous emphysema, pneumothorax and pneumomediastinum, it had also been reported by the radiologist, and, conversely, the assessors never found subcutaneous emphysema, pneumothorax or pneumomediastinum in chest images in which the radiologists did not report an air leak. Consequently, these findings did not reveal a difference between CXR and chest CT scans.

In two patients (1.3%), an air leak was present at ICU admission and, thus, before the start of invasive ventilation. Air leaks that developed during the patients’ stay in the ICU became visible at a median of 18 [2 to 21] days after ICU admission and at a median of 18 [9 to 22] days after start of invasive ventilation (Figure 3). None of the air leaks was judged to be related to a central venous catheter placement or tracheotomy. Of eight patients with a pneumothorax, four were treated with a chest tube, and in two patients there was mention of a tension pneumothorax.

### 3.3. Follow-Up

Patients with an air leak had a lower 28-day mortality rate, but a higher ICU mortality rate, than patients without an air leak. It is of note that patients with an air leak received ventilation for a longer time and also stayed in the ICU longer (Table 3 and Appendix A).

## 4. Discussion

The findings of this preplanned analysis of two cohorts of patients with acute hypoxemic respiratory failure due to COVID-19 can be summarized as follows: (1) air leaks occurred often in these patients and included subcutaneous emphysema, pneumothorax and pneumomediastinum; (2) all air leaks occurred in patients receiving ventilation, and the majority occurred rather late in the course of the ICU stay; (3) patients with air leaks were comparable to patients without air leaks with respect to demographics and baseline characteristics; and (4) half of the patients with pneumothoraxes were treated with a chest tube.

This study has several strengths. To ensure the high quality of the data obtained, data collectors were trained and provided with strict instructions before starting to capture data. The parent studies had no exclusion criteria, other than patients having an alternate diagnosis. For the current analysis, we only excluded patients on whom no CXR or chest CT scan was performed. All available chest images were rescored by three independent investigators and, if needed, an independent radiologist, in order to come to consensus, improving the validity of our findings. Moreover, clear definitions were used for scoring the presence or absence of air leaks. Follow-up was near to complete, the amount of missing data was acceptable, and the analysis strictly followed a predefined analysis plan.

The findings of our study are not in line with previous reports on air leaks in patients with acute hypoxemic respiratory failure due to COVID-19 [11,13,16,17,18,19,20,21,22,23]. The proportion of patients with air leaks in our study was remarkably lower than the reported incidences in nearly all published studies [11,13,16,17,18,19,21,22,23]. In fact, only one study reported a lower incidence [20]. Indeed, the reported incidences in previous studies detailing a higher rate of incidence varied from 9.6% to as high as 24.4%, with a pooled incidence of 15.6% [9]. This difference cannot be explained by differences in patient selection or by the exclusion of patients from our study, e.g., because of a missing CXR or chest CT scan. The distribution of types of air leaks, however, was similar to what was reported before, with pneumothorax being the most common air leak [9]. Even the preferred location of pneumothoraxes was similar to what was reported in previous studies [9].

Air leak incidences in our study, however, are much more in line with the reports on air leaks in the pre-COVID-19 era, i.e., in patients with acute hypoxemic respiratory failure due to a cause other than COVID-19 [3,6,7,24,25]. Indeed, in those studies, the incidence of air leaks ranged from 5 to 13%. What is different, though, is the timing of the air leaks. In our cohort of COVID-19 patients, as in the other studies on COVID-19, air leaks occurred much later than in the studies originating from before COVID-19 [3,24]. The reason for this remains unclear.

One meta-analysis, including 15 studies, suggests a linear relation between disease severity and the development of air leaks [9]. Patients with an air leak in our cohort were ventilated with a higher median of FiO_2_ than patients without an air leak. This could mean that the lungs of patients who developed an air leak were more affected than those of patients who did not develop this complication. There were no other differences in ventilator settings between patients with air leaks and patients without air leaks in our cohort. Incidences of air leaks may be driven, at least in part, by how ventilators are set [26]. Indeed, ventilation with a low VT or low airway pressures may prevent gross barotrauma in patients with ARDS; prone ventilation may also have a protective effect, as it could facilitate the use of lower airway pressures [5]. In our cohort, the VT, PEEP and Pmax were within widely accepted safety limits, and prone position was also heavily used. One salient finding was that the median peak airway pressures and PEEPs in our cohort were lower than what was reported in the previous studies with a higher incidence of air leaks [19,21]. These finding suggest that differences in air leak incidences between studies could have been driven by ventilator settings.

In line with one meta-analysis that showed a higher in-hospital mortality in patients with air leaks [9], we found a higher rate of ICU mortalities in patients with air leaks. However, our finding was that the 28-day mortality rate was lower in patients with air leaks. This may seem contradictory but can be explained by longer lengths of ICU stays in this group. The finding that the ICU and hospital lengths of stay were longer for patients with an air leak than patients without an air leak could be explained by the fact that the former may have been sicker. One could also hypothesize, however, that a longer exposure to mechanical ventilation and an increased pulmonary inflammatory response increases the risk of the development of air leaks [27]. The fact that we found air leaks late in the ICU trajectory supports this hypothesis.

Our findings are important for clinical practice in several ways. Healthcare professionals should be aware of the considerable risk of air leaks in the ICU and, especially, in critically ill patients ventilated for acute hypoxemic respiratory failure due to COVID-19. However, our results suggest that the risk for patients with COVID-19 is approximately the same as for patients with ARDS not caused by COVID-19. Differences in ventilator settings between studies with high and low incidences of air leaks suggest potentially modifiable risk factors.

Our study has limitations. This was a single-center study, limiting the generalizability of our findings. Second, as we assessed imaging up to day 28, we cannot exclude the possibility of a higher incidence if we had extended the observation period. Last but not least, the number of patients receiving HFNO was small, making any comparison between the modes of respiratory support at risk of type II errors.

## 5. Conclusions

The incidence of air leaks in this cohort of COVID-19 patients with acute hypoxemic respiratory failure was high, but it was not different from that observed in cohorts of patients with acute hypoxemic respiratory failure due to other causes. Half of the patients with a pneumothorax were treated with a chest tube.

## Figures and Tables

**Figure 1 diagnostics-13-01156-f001:**
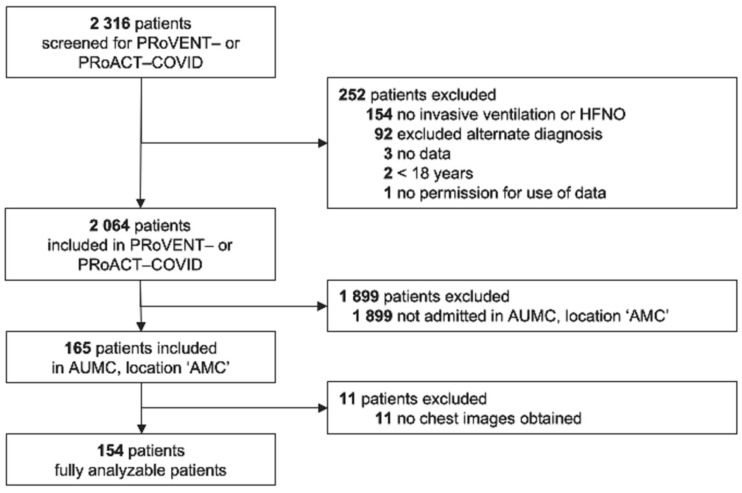
CONSORT diagram. Patient flow in the parent studies and for this analysis. Abbreviations: PRoVENT–COVID—Practice of Ventilation in COVID-19 patients; PRoAcT–COVID—Practice of Adjunctive Therapies in COVID-19 study; HFNO—high-flow nasal oxygen; UAMC—Amsterdam University Medical Centers; and AMC—Academic Medical Center.

**Figure 2 diagnostics-13-01156-f002:**
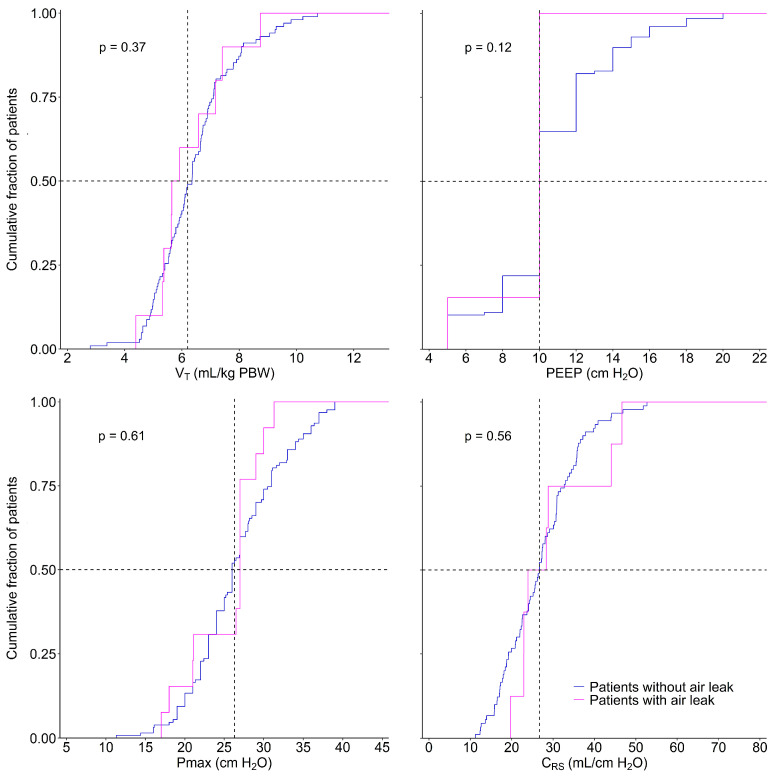
Key ventilation parameters. Cumulative frequency distribution of key ventilation parameters. Vertical dotted lines represent the median for each variable; horizontal dotted lines show the respective proportion of patients reaching the median. Abbreviations: VT—tidal volume; PEEP—positive end-expiratory pressure; Pmax—maximum airway pressure; and CRS—respiratory system compliance.

**Figure 3 diagnostics-13-01156-f003:**
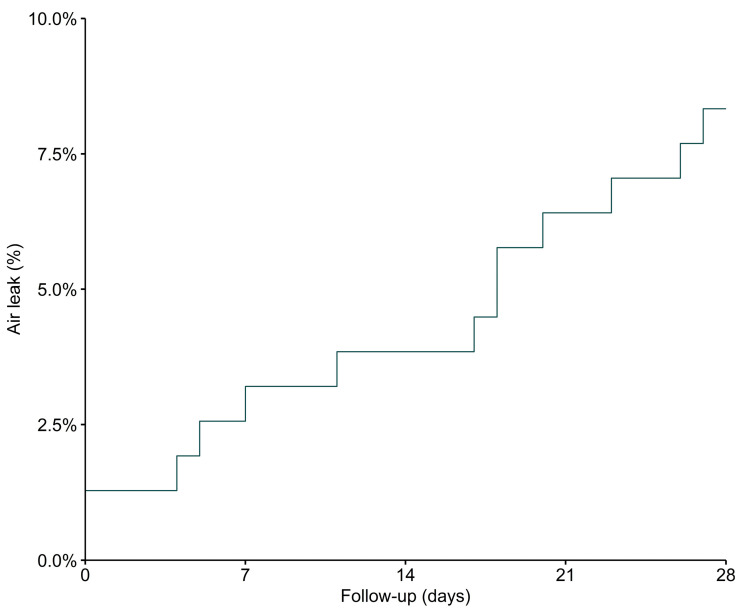
Time to first confirmation of an air leak. Time from ICU admission to first detection of an air leak on CXR or chest CT scan.

**Table 1 diagnostics-13-01156-t001:** Demographics and Baseline Characteristics.

	Patients with Air Leak (N = 13)	Patients without Air Leak (N = 143)	* p *
Age, years	63.0 [56.0–72.0]	62.0 [55.0–70.5]	0.92
Sex, male	11/13 (84.6)	103/143 (72.0)	0.52
BMI, kg/m^2^	26.9 [23.1–29.4]	28.7 [25.4–32.3]	0.22
SOFA score	6.0 [5.0–8.0]	6.0 [5.0–8.0]	0.92
Medical history, n/N (%)			
COPD	1/13 (7.7)	11/143 (7.7)	1.00
Asthma	1/13 (7.7)	13/143 (9.1)	1.00
Lung emphysema	0/13 (0.0)	3/143 (2.1)	1.00
Interstitial lung disease	0/13 (0.0)	1/143 (0.7)	1.00
Active smoker	1/6 (16.7)	4/77 (5.2)	0.81
History of smoking	3/5 (60.0)	33/74 (44.6)	0.84
Previous pneumothorax	0/13 (0.0)	0/143 (0.0)	
Home medication, n/N (%)			
Ace inhibitors	2/12 (16.7)	22/115 (19.1)	1.00
Angiotensin receptor blocker	1/12 (8.3)	15/115 (13.0)	1.00
Ventilatory support			
with invasive ventilation, n/N (%)	13/13 (100.0)	133/143 (93.0)	0.69
FiO_2_, %	80.0 [65.0–90.0]	60.0 [45.0–75.0]	0.02
V_T_ mL/kg PBW	5.6 [5.4–6.9]	6.4 [5.4–7.1]	0.37
PEEP, cm H_2_O	10.0 [10.0-10.0]	10.0 [10.0–12.0]	0.12
Pmax, cm H_2_O	27.0 [21.1–27.0]	26.0 [23.0–31.0]	0.61
C_RS_, mL/cmH_2_O	26.2 [22.9–32.6]	26.7 [19.6–32.9]	0.56
Prone positioning	12/13 (92.3)	120/133 (90.2)	1.00
with high flow nasal oxygen	0.0 (0.0)	10/143 (7.0)	0.69
FiO_2_, %	N.A.	75.0 [60.0–80.0]	
Flow, L O_2_/min	N.A.	50.0 [41.3–57.5]	
Prone positioning	N.A.	3/10 (30.0)	
Horowitz index			
PaO_2_/FiO_2_	115.0 [80.6–126.8]	123.8 [90.8–175.5]	0.14
PaO_2_/FiO_2_ > 300 mmHg	0/13 (0.0)	6/141 (4.3)	
PaO2/FiO_2_ 200–300 mmHg	0/13 (0.0)	14/141 (9.9)	
PaO_2_/FiO_2_ 100–200 mmHg	7/13 (53.8)	77/141 (54.6)	
PaO_2_/FiO_2_ < 100 mmHg	6/13 (46.2)	44/141 (31.2)	

Data are n/N (%) or median [25–75% quartile]; proportions may not total 100 because of rounding. Abbreviations: Body Mass Index—BMI;SOFA—Sequential Organ Failure Assessment; COPD—chronic obstructive pulmonary disease; F_i_O_2_—Fraction of inspired oxygen; V_T_—tidal volume; PEEP—positive end-expiratory pressure; Pmax—maximum airway pressure; C_RS_—respiratory system compliance; PC—pressure controlled; P_a_O_2_—partial arterial oxygen pressure.

**Table 2 diagnostics-13-01156-t002:** Air Leak Characteristics in Patients with an Air Leak.

Number of Patients	N = 13
Pneumothorax	8/13 (61.5)
Left	1/8 (12.5)
Right	6/8 (75.0)
Bilateral	1/8 (12.5)
Pneumomediastinum	6/13 (46.2)
Apical alone	0/6 (0.0)
Basal alone	0/6 (0.0)
Apical + basal	6/6 (100.0)
Subcutaneous emphysema	7/13 (53.8)
Patients with a combination of air leaks	5/13 (38.5)
Pneumothorax + Subcutaneous emphysema + Pneumomediastinum	3/5 (23.1)
Pneumothorax + Pneumomediastinum	0/5 (0.0)
Pneumothorax + Subcutaneous emphysema	1/5 (7.7)
Pneumomediastinum + Subcutaneous emphysema	1/5 (7.7)
Patients with an isolated air leak	
Pneumothorax	4/8 (50.0)
Pneumomediastinum alone	2/8 (25.0)
Subcutaneous emphysema alone	2/8 (25.0)

Data are n/N (%).

**Table 3 diagnostics-13-01156-t003:** Clinical Outcomes of Patients Categorized According to Presence of Air Leak.

	Patients with Air Leak (N = 13)	Patients without Air Leak (N = 143)	* p *
Duration of respiratory support, days	20.0 [12.0–28.0]	10.0 [6.0–17.8]	0.05
ICU length of stay, days	40.0 [12.0–44.0]	11.0 [7.0–18.0]	0.004
ICU mortality	6/13 (46.2)	50/106 (35.0)	0.62
28-day mortality	3/13 (23.1)	52/143 (36.4)	0.51
90-day mortality	2/13 (15.4)	25/143 (17.5)	1.00

Data are n/N (%) or median [25–75% quartile]; proportions may not total 100 because of rounding.

## Data Availability

The data presented in this study are available on request from the corresponding author. The data are not publicly available due to privacy concerns.

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
