# Peer review of "Incidence of Air Leaks in Critically Ill Patients with Acute Hypoxemic Respiratory Failure Due to COVID-19"

_diagnostics, 2023, doi:10.3390/diagnostics13061156_

Round 1

Reviewer 1 Report

Major

It is true that Intensive care unit (ICU) patients, and mechanically ventilated critically ill patients in particular are at risk of air leaks. Although many patients with COVID-19 pneumonia were treated with mechanical ventilation under COVID-19 pandemic, the incidence of air leaks in critically ill COVID–19 patients have not been elucidated.

The current authors have examined the incidence of subcutaneous emphysema, pneumothorax and pneumomediastinum in critically ill patients with acute hypoxemic respiratory failure due to COVID–19. They found that the incidence of subcutaneous emphysema, pneumothorax and pneumomediastinum was 4.5%, 4.5%, and 3.9%, present in 13 (8.4%) patients. The median first day of presence of an air leak was 18 days after arrival in the ICU, and 18 days after start of invasive ventilation. The incidence of air leaks was high in this cohort of COVID–19 patients, but comparable with what was previously reported in patients with acute hypoxemic respiratory failure in the pre–COVID–19 era. The incidence of air leak complication with invasive ventilation may depend on the management of mechanical ventilation mode and low tidal volume theory. Therefore the data may be affected by the experiences of ICU staff and lung protective strategy. The other important issue of invasive ventilation for COVID-19 is prone position approach.   It is well established that prone positioning reduces mortality from acute respiratory distress syndrome in the low tidal volume era(Intensive Care Med 40:332-341, 2014). The influences of prone positioning on the incidence of air leaks in patients with invasive ventilation due to COVID-19 should be discussed.  

Reviewer 2 Report

A job is well done and well written. The goal is concrete – simple. The results and conclusions are clear.

The work includes a very practical "technical" issue – the detection of air (pneumothorax and pneumomediastinum) on chest x-ray images (it is usually easy to detect on CT images). Therefore, it would be interesting and important if the authors indicated in the text if were cases of pneumothorax and/or pneumomediastinum detected only retrospectively during this study, i.e. overlooked before.

It would also be useful if the authors indicated what percentage of agreement/disagreement there was between study authors who evaluated the presence of pneumothorax and pneumomediastinum on chest x-ray images (Lines 127-131).
